# Rim Breakups of Impacting Drops on a Superhydrophobic Surface and a Superheated Surface

**Minori Shirota \*, Masaki Kato and Ai Ishio**

Graduate School of Science and Technology, Hirosaki University, 3 Bunkyocho, Hirosaki 0368561, Aomori, Japan; masaking1904@hotmail.co.jp (M.K.); ishio071218@outlook.jp (A.I.)
\* Correspondence: mshirota@hirosaki-u.ac.jp

**Abstract:** The rim breakup of an impacting drop is experimentally investigated by comparing the impacts on superheated and superhydrophobic surfaces. The objective of the present study is to experimentally examine whether the $Bo = 1$ criteria holds for the rim breakups of drops impacting on the surfaces. A transparent sapphire plate was heated to achieve the Leidenfrost impact, which enables us to observe with a high-speed camera from below. The characteristics of the rim breakup were evaluated quantitatively using a particle tracking velocimetry method for both the rim and the drops generated. As a result, we clarified that $Bo$ of the rim increases in the spreading phase and marks the highest value of 0.5 on a superheated surface, which is smaller than that on a pillar, where $Bo \approx 1$. On a superhydrophobic surface, the highest $Bo$ was 1.2, which is smaller than that on a wettable solid surface, 2.5, but close to the value on a pillar. We also revealed that diameters of generated drops collapse on a master curve when plotted as a function of pinch-off time for both the impacts on superheated and superhydrophobic surfaces.

**Keywords:** drop impact; Leidenfrost; hydrophobic; rim breakup; unsteady atomization





## 1. Introduction

Drop impact is an important fundamental process that determines the quality and the efficiency in many industrial applications including painting, coating, engine combustion, and spray cooling. Therefore, the physics of drop impact have been widely investigated. Among them, the maximum spreading diameter has been studied in detail. Many theoretical models on the maximum spreading have been derived based on the conservation of energy for the impact of a pancake-shaped drop [1–9]. More realistic models for the shape of the impacting drop were also derived [10]. Furthermore, expanding and contracting dynamics of the rim are also a target in the study of drop impact [5,10–12].

The number of studies about the drop impact with phase change, such as boiling and solidification, have rapidly increased in the past 10 years. In the spray cooling, nucleate boiling plays a crucial role to achieve the critical heat flux, while higher substrate temperatures or lower ambient pressures result in the dynamic Leidenfrost [13–18], which should be avoided in the cooling application. Solidification of the impacting metal drop controls the characteristics of mechanical strength, and electrical and heat transfer rates of the products manufactured by metal 3D printing [19–21].

The rim breakup of an impacting drop is sometimes referred to as unsteady atomization [22,23]. It is called unsteady because the diameter of the rim, the ligament, the generated drop, and the acceleration of the film are all functions of time. As a result, the drop size distribution shows multi-dispersed ones, although the drops are generated by primary breakup process only.

Unsteady atomization has been studied mainly with drop impacts on a pillar [11,22,24]. The prior studies showed that the momentum balance reads the rim diameter $b$ scales as

capillary length with the acceleration of the film surrounded by the rim, $\ddot{R}$. This condition can be described in dimensionless form as

$$Bo = \frac{\rho(-\ddot{R})b^2}{\sigma} \simeq 1, \tag{1}$$

where $\rho$ and $\sigma$ are the liquid density and the surface tension coefficient. The condition $Bo = 1$ matches the experimental results very well on a pillar, but not on a flat surface nor on a free surface of a deep pool [23]. On the flat surface, where the viscous friction slows down the spreading motion, the $Bo$ gradually increases and marks the highest value of about 2.5 during the spreading, while on the free surface $Bo \approx 0.5$ all the spreading time.

In summary, for the rim breakup, we have the critical $Bo$ of 1 on a pillar, and larger than 1 on a solid large surface probably due to the viscous friction on the wall. This consideration leads us to a guess for the other limit, Leidenfrost state on a superheated (SH) surface. In the Leidenfrost state, impacting drops spread by hovering on their own vapor. Therefore, the vapor flow might enhance the rim breakup and thus the critical $Bo$ might become smaller than one. Another situation where a spreading rim does not contact the substrate can be achieved on a superhydrophobic (SHPB) surface.

The objective of the present study is to experimentally examine whether or not the $Bo = 1$ criteria holds for the rim breakups of drops impacting on a superhydrophobic surface and a superheated one above the Leidenfrost temperature. To quantitatively evaluate the characteristics of the rim breakup, we employed high-speed imaging and image analysis based on particle tracking velocimetry (PTV) method.

## 2. Materials and Methods

In experiment, single water drops of 2.8 mm and ethanol drops of 1.6 mm in diameter were released from a needle to a substrate. We set the needle height at about 0.2 m so that the drop impact velocity was kept constant at 2.0 m/s. The corresponding Weber numbers were about 160 and 250 for water and ethanol, respectively.

The substrate was a circular sapphire plate having the thickness of 5.0 mm and the diameter of 50.0 mm. The sapphire substrate enables us not only to observe from the bottom, but also to achieve the Leidenfrost state even for water drop due to its relatively high thermal conductivity and the tolerance for the thermal shock. In order to highlight the vapor effect on the rim breakup in the Leidenfrost state, drop impact on the sapphire substrate with a superhydrophobic coating (Glaco, Soft99 Corporation, Osaka, Japan) was also examined. The surface temperature of the substrate was PID controlled with an electrical heater to be 410 °C to achieve the Leidenfrost state, while the SHPB surface was set at the room temperature.

The drop impact processes were observed using a high-speed camera (Fastcam SA-Z, Photron Limited, Tokyo, Japan) equipped with a long working distance microscope (Z16-APO, Leica Camera Japan Co., Ltd., Tokyo, Japan), as shown in Figure 1. A key trick to clearly observe the rim breakups is the diffuser rotating with a stepping motor. When a drop is about to impact the substrate, the laser sensor triggers the stepping motor to rotate a tracing paper and diffuse the back light. In this way, the diffuser did not interrupt a drop from falling from the needle.

The instantaneous and local geometrical characteristics related to the rim dynamics of impacting drops were quantitatively determined through image analysis using MATLAB. The final outcome that we want to quantitatively evaluate after the image processing is the time-dependent $Bo$. To obtain $Bo$, we need to know the acceleration and the diameter of the rim as well as the physical properties, the surface tension, and the density.

Here, the difficulty arises in the image processing, as the rim diameter varies not only temporarily but also spatially in the azimuthal direction due to the instability. We measured the change in the local thickness, or the curvature, of the rim in the azimuthal direction by filling the rim with circles that have the centers along the centerline of the rim and touch the interface of the rim.

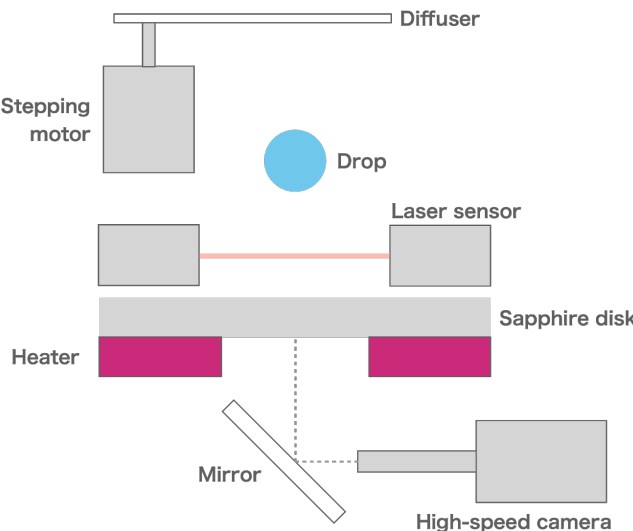

**Figure 1.** Schematic diagram of experimental setup.

The image processing took the following steps. Firstly, an original image (Figure 2a) was binarized and any holes within the rim and the fingers were filled (Figure 2b). From the binarized image, center lines, or the so-called skeleton image, of both the rim and finger were obtained after several expansion and contraction morphological processes (Figure 2c). Next, the centerline of the rim was distinguished from those of the fingers based on the idea that the centerline of the rim was the only closed line and had the longest length. Finally, the local curvatures of the rim and the fingers were determined by fitting circles which have the center along the centerline and the diameter that touches the edges of the binarized image (Figure 2d). In Figure 2e, we superimposed the circles to the original image, which clearly shows that the local thicknesses of both the rim and finger are well described by the circles and are well distinguished from each other. From the variation data of the local rim thickness in the azimuthal direction, i.e., the variation in the diameter of green circles in Figure 2e, we obtained the median of the rim thickness and the sheet radius, $b_{\mathrm{med}}(t)$ and $R_{\mathrm{med}}(t)$, respectively.

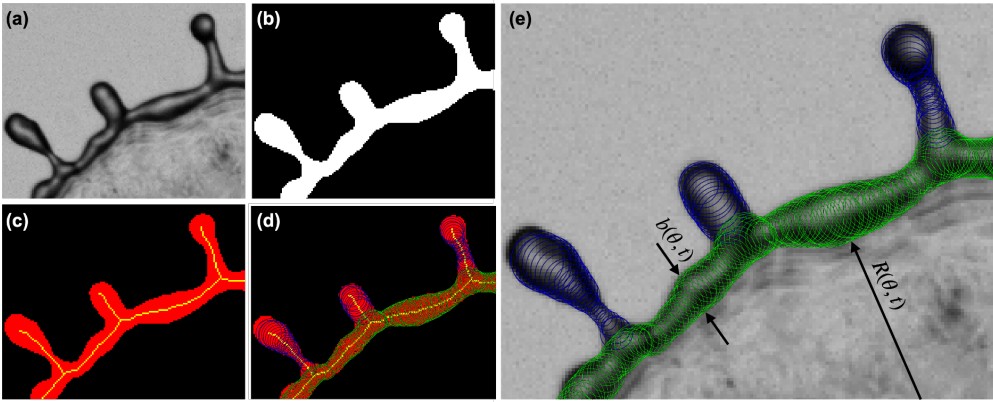

**Figure 2.** Image processing to obtain local thickness of the rim and the ligament. (**a**) Original image, (**b**) binarized image, (**c**) center line, (**d**) circles having the centers along the centerline and radii to fit to the boundary, (**e**) resultant circles fitted to the rim (green) and the fingers (blue).

We then determined the instantaneous rim diameter, $b(t)$, by taking the arithmetic average of the median of the variations in azimuthal direction, $b_{\mathrm{med},i}(t)$,

$$b(t) = \frac{1}{N_d} \sum_{i=1}^{N_d} b_{\mathrm{med},i}(t), \tag{2}$$

where the number of mother drops, $N_d$, was 10 for water and 11 for ethanol drops for each substrate condition. The radius of the film, $R(t)$, surrounded by the rim was also determined in the same manner as $b(t)$,

$$R(t) = \frac{1}{N_d} \sum_{i=1}^{N_d} R_{\mathrm{med},i}(t), \tag{3}$$

where $R_{\mathrm{med},i}(t)$ is the median of the variations in the distance from the impaction point to the inner edge of the rim at time $t$ of the $i$th mother drop. The instantaneous acceleration of the spreading film, $\ddot{R}(t)$, required for the calculation of $Bo$ (Equation (1)) was obtained from the coefficients of a fourth-order polynomial function that fits to $R(t)$.

After obtaining the area and positions of drops separated from the rim in all the frames, we linked the identical drops in consecutive frames by using the PTV method. Figure 3 illustrates the concept of linking drops between frame $j$ and $j + 1$. We first filtered out drops whose area is either smaller than $0.1S$ or greater than $1.9S$ in the search region with $S$ being the area of the drop in frame $j$. We then determined that the target drop in frame $j + 1$ is the one that has the closest distance from the center of the search region. Here, the center of the search region, $r_s$, was given by the position vectors of the same drops in the present and the last frame, i.e., $r_s = r_j + (r_j - r_{j-1}.) = 2r_j - r_{j-1}$. The search region has some extensions in radial and azimuthal directions, $\Delta r$ and $\Delta \theta$; see Figure 3a.

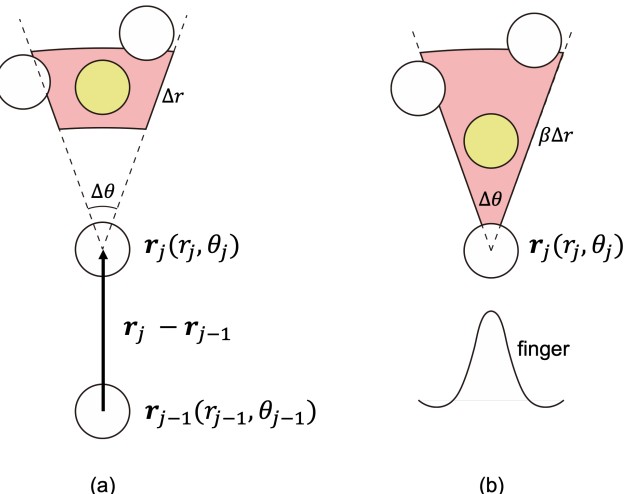

(a)                                (b)

**Figure 3.** Schematic diagram showing how to find the identical drop in the next time step. The red region and the yellow circle represent the search region and the target drop, respectively, (**a**) during daughter drops are moving and (**b**) immediately after the pinch-off.

This numbering method works for any drops except the ones immediately after the pinch-off, which do not have $r_{j-1}$. In this case of the very initial moment, we determined that the target drop in frame $j + 1$ is the one that has the closest area, or equivalent diameter, to the same drop in frame $j$ within a search region. Here, the search region is defined as a fan-shaped area having the vertex at the center of the $j$-th drop and the center line along $\theta_j$ direction, as shown in Figure 3b. The radial extension length of the search region is $\beta$ times larger in case (b) than (a), where $\beta(>1)$ is a parameter adjusted depending on the impact velocity and the substrate.

We proceeded the drop tracking method in the forward direction in time. In a new frame, we first found the drops satisfying the condition (a) in Figure 3. Then, for the remaining drops, i.e., the drops satisfying the condition (b) in Figure 3, we determined the pinch-off time of the drop from the frame number.

## 3. Results

In Figure 4, we compare the impact dynamics on a superhydrophobic surface in the left-half panel and on a superheated surface in the right-half panel. Note that the initial drop diameter and the impact velocity are the same for the two cases. However, the time variation in the global rim dynamics differ from each other. On the superheated surface, the rim spread faster and reached a larger maximum diameter. After the maximum expansion, the rim shrunk on the SHPB surface due to the surface tension of the film surrounded by the rim, while on the SH surface, both the rim and film disintegrated into small pieces.

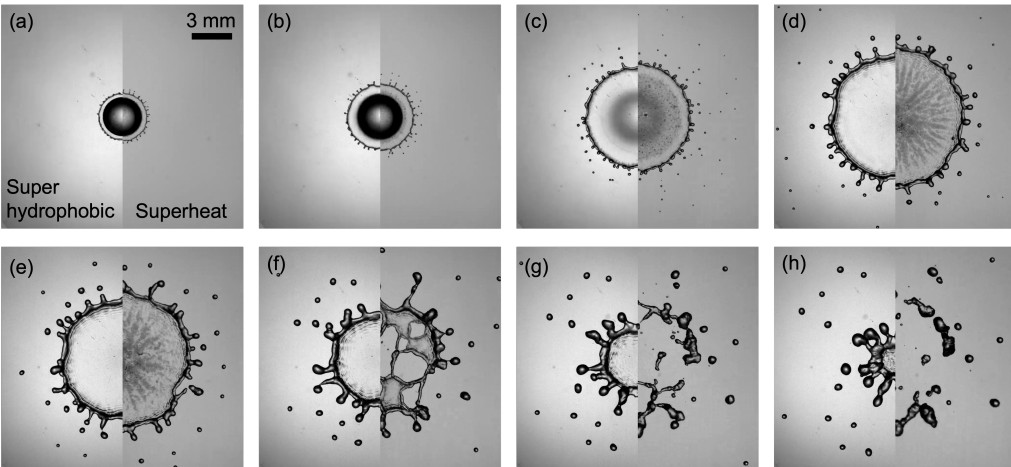

**Figure 4.** Comparison of drop impact between on (**left**) a superhydrophobic surface and (**right**) in Leidenfrost state. The elapsed time from the contact is (**a**) 0.25 ms, (**b**) 0.4 ms, (**c**) 1.0 ms, (**d**) 2.5 ms, (**e**) 3.5 ms, (**f**) 5.0 ms, (**g**) 6.0 ms, and (**h**) 7.0 ms.

The diameters of the rims increase with time on both surfaces. A closer observation reveals that the rim on the SH surface has smaller diameter than on SHPB. Another difference appears on the rim fingering characteristics: On the SH surface, the fingers elongate faster, leading to the earlier pinch-off of smaller daughter drops. We quantitatively evaluated the characteristic length scales of the rim, fingers, and daughter drops by using PTV. The results will be shown in the following subsections.

### 3.1. Local Bo of the Rim

The bottom view images in Figure 4 clearly show that the rim diameter increases with time on both the substrates. We will then compare the rim diameter with the instantaneous capillary length. Note that in the drop impact on a pillar, the rim does not exceed an instantaneous capillary length [22]. In other words, in terms of *Bo*, the condition *Bo* = 1 holds all the time on a pillar. For the impact on SH and SHPB surfaces, we quantitatively evaluated the rim thickness, $b(t)$, with the home-brewed image processing described in Section 2. The time change in the radius of expanding sheet, $R(t)$, surrounded by the rim was also obtained to determine the local acceleration of the rim, $\ddot{R}(t)$.

Figure 5 shows that $b(t)$ is smaller and $R(t)$ is larger in Leidenfrost state than on SHPB surface. $\ddot{R}(t)$ shown in Figure 5b crosses during the spreading phase. The error bars on Figure 5a represent the standard deviation of ten impacts. The larger error bars for Leidenfrost state reflect the poor reproducibility due to the local contact of the spreading film caused by the partial cooling of the sapphire substrate [16]. The local contacts are visualized in the right panels of Figure 5c,d as dark spots.

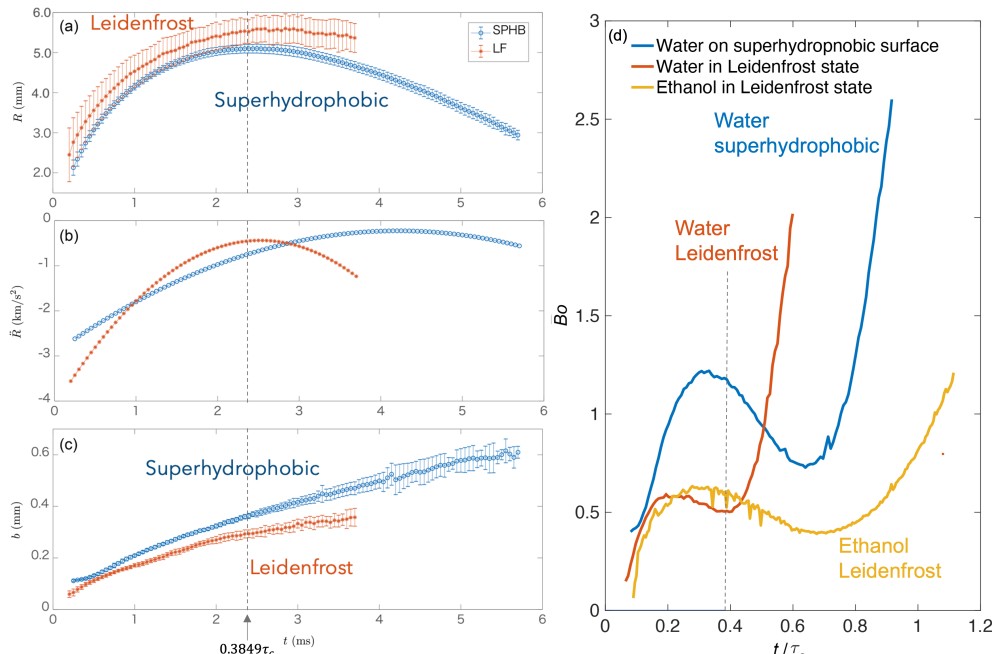

**Figure 5.** Time variation in (**a**) the radius of the spreading film $R(t)$, (**b**) the acceleration of the spreading film $\ddot{R}(t)$, and (**c**) the diameter of the rim $b(t)$. The resultant $Bo$ is shown in (**d**) as a function of dimensionless time $t/\tau_c$, with $\tau_c = \sqrt{\rho D_0^3/8\sigma}$ being the capillary time of the impacting drop. $\tau_c = 6.2$ ms for water drop and 4.5 ms for ethanol drop. The dashed lines represent, for water drop, $t = 0.3849\tau_c = \frac{1}{3}\sqrt{\rho D_0^3/6\sigma}$, which is the measure for the maximum spreading time of impacting drops on a pillar [11]. Note that the plots of $R(t)$ and $b(t)$ show the average value in both the azimuthal direction and over ten experimental runs, and the error bars show the standard deviation of ten experiments; see Equations (2) and (3).

Figure 5d reveals that, before the maximum spreading, $Bo \approx 0.5$ in the Leidenfrost state for both water and ethanol, while $Bo$ marks 1.2 on the SHPB surface. In both cases, $Bo$s are not constant but increase with time and slightly decrease during the spreading phase. In the following contracting phase, $Bo$ increases again in both cases due to the rim thickening.

By comparing with the impact on a solid surface at room temperature with no surface modification [23], we clarify that the increasing behavior of $Bo$ in the spreading phase was reproduced here on both the SH and SHBP surfaces. However, they are different in quantitative manner: On the normal surface, the maximum $Bo$ is 2.5, while on SHPB surface it decreases to 1.2. On SH surface, the $Bo$ further decreases to 0.5, which is almost the same as the one obtained on a free surface of a deep pool.

### 3.2. Pinch-Off of the Daughter Drops from the Rim

For generated drops, we applied the PTV method to evaluate the drop diameter, velocity, pinch-off time, pinch-off position, and, more importantly, to identify every drop so that we can avoid double counting in statistical analysis.

Examples of the PTV result are shown in Figure 6 for (a) Leidenfrost state and (b) on SHPB surface. Note that all frames in the initial stage of an impact, taken at 20,000 fps, are superimposed. Outlines of the generated drops are shown by the circles. The color of the circles changes based on the pinch-off order from blue to red, as shown by the color contours. Note that several typical times are also shown in ms on the top by triangles to estimate the pinch-off time.

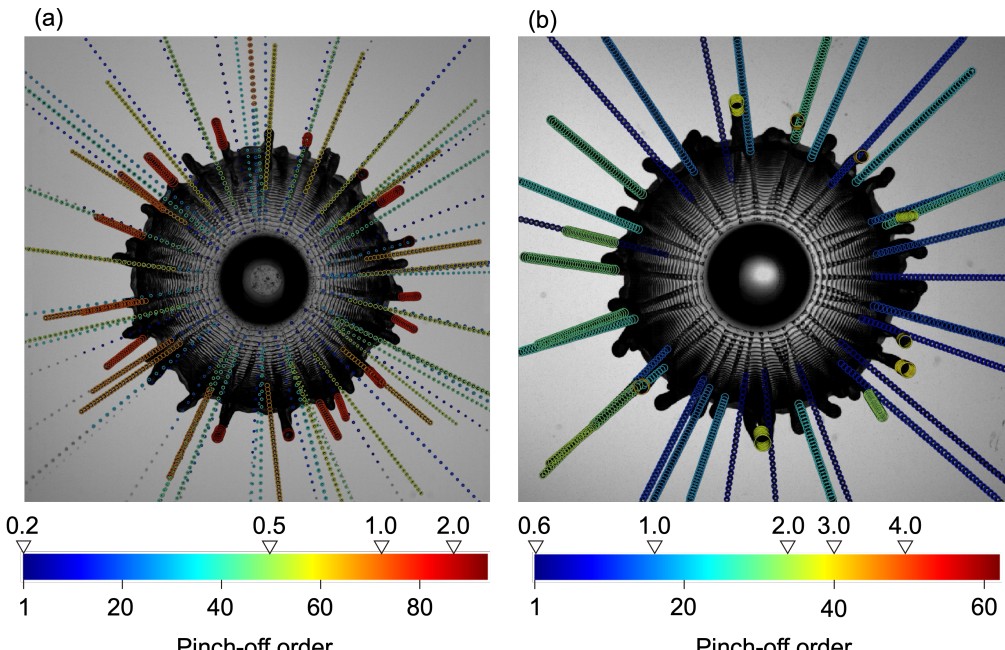

**Figure 6.** Superimposed images of drop impact on a (**a**) superheated and (**b**) superhydrophobic surfaces. The colors of daughter drops show the pinch-off order as shown by each contour. Typical pinch-off times are also shown in ms on the top of contour by triangles. Only the expanding phase is shown here. For the pinch-off in later stage, see Videos S1 and S2 provided in Supplementary Materials.

From Figure 6a, we see, in Leidenfrost state, smaller drops are generated and move away at faster and constant velocities, as described by the constant spacing in the consecutive frames. All the daughter drops flights are on the paths pointing radially outward from the center of the mother drop, with some exceptions originating from merged neighboring fingers in the later stage. On SHPB surface, shown in Figure 6b, the daughter drops are released later and larger. The diameters of drops shown in blue colors in (b) almost correspond to orange or even red ones in (a). It is also noteworthy that on SPHB surface, the spacings between the drops in consecutive frames are shorter, indicating slower flight velocities.

Therefore, the drop size distributions differ between the two cases, as shown in Figure 7. The mode diameter on SHPB is 0.2 mm, while it is about 0.1 mm in Leidenfrost state. The particle size distributions show the skewed distribution to the lower diameter-side on both the surfaces, which is the characteristic of the drop breakup well described by a gamma function [22], supporting the aggregation scenario for the breakup [25,26].

However, when the diameter data in Figure 7 are plotted as a function of pinch-off time, all the data surprisingly collapse on a single curve, as shown in Figure 8. The log–log plot of the daughter drop diameter, $d(t)$, shown in Figure 8b, reveals that $d \sim t_p^{0.68}$, where $t_p$ represents the pinch-off time. The scaling was originally derived in Visser et al. [27] not for the daughter drop diameter, $d(t)$, but for the rim diameter, $b(t)$, of impacting drops. The coincidence in the scaling law between $d(t)$ and $b(t)$ suggests $d(t) \propto b(t)$, which will be proved in the following. Note here that the thickness of the rim, $b(t)$, from which the daughter drops pinch off follows the different time evolution on SH and SHPB surfaces, as already shown in Figure 5.

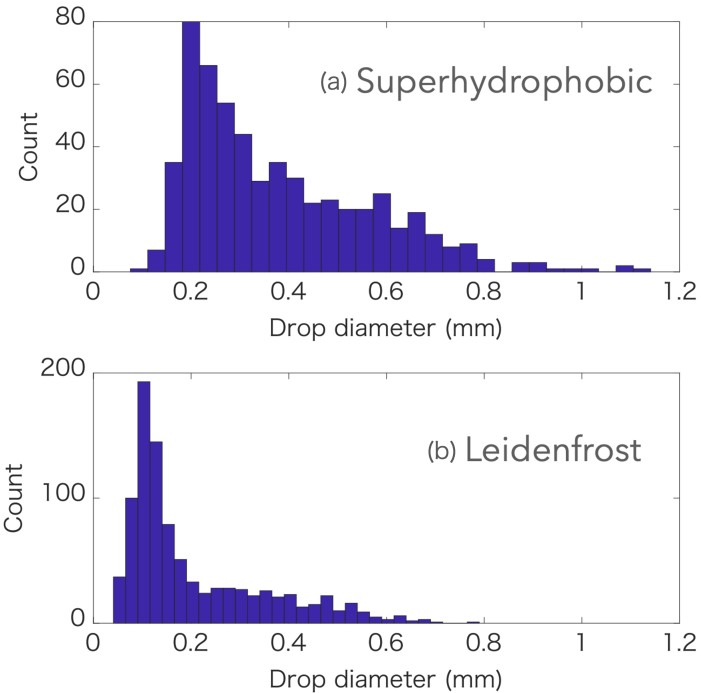

**Figure 7.** Histograms of daughter drop diameters generated on a (**a**) superhydrophobic and (**b**) superheated surface.

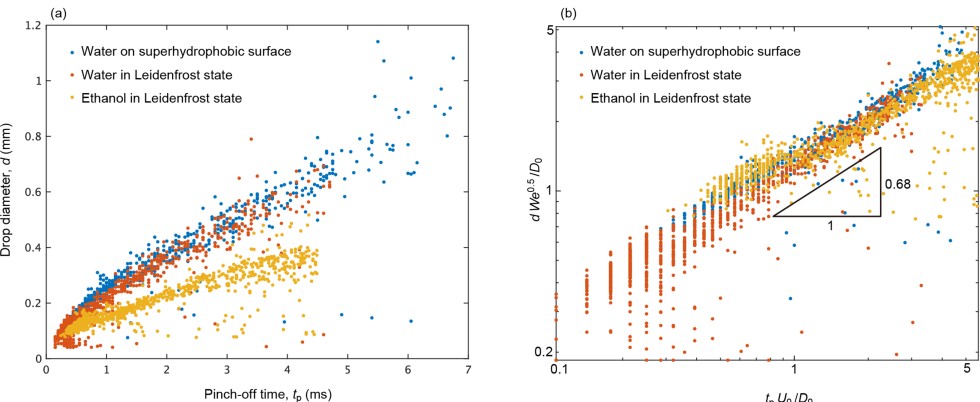

**Figure 8.** Diameters of daughter drops plotted as a function of pinch-off time, $t_p$, from the rim expanding on a superhydrophobic or superheated surface above Leidenfrost point. The same data are shown on (**a**) linear axes with the dimensional ones and (**b**) double logarithmic in dimensionless forms. The characteristic length scale, $D_0/We^{0.5}$, in (**b**) is the measure of the maximum spreading diameter of the mother drop [11].

The collapse on a function of pinch-off time (Figure 8) for datasets that differ in drop size distribution (Figure 7) suggests that the daughter drops pinch off the mother drop earlier in Leidenfrost state than on SPHB surface. This is confirmed with the histogram for the pinch-off time shown in Figure 9. The figure also shows an interesting characteristic of the rim breakup where the pinch-off frequency decreases not monotonically but with several peaks, as shown by the arrows, which is the characteristic of the unsteady atomization where drops pinch off when the local *Bo* exceeds a critical value [22,23]. However, as shown in the previous section, the critical *Bo*s differ on SH and on SHPB surfaces.

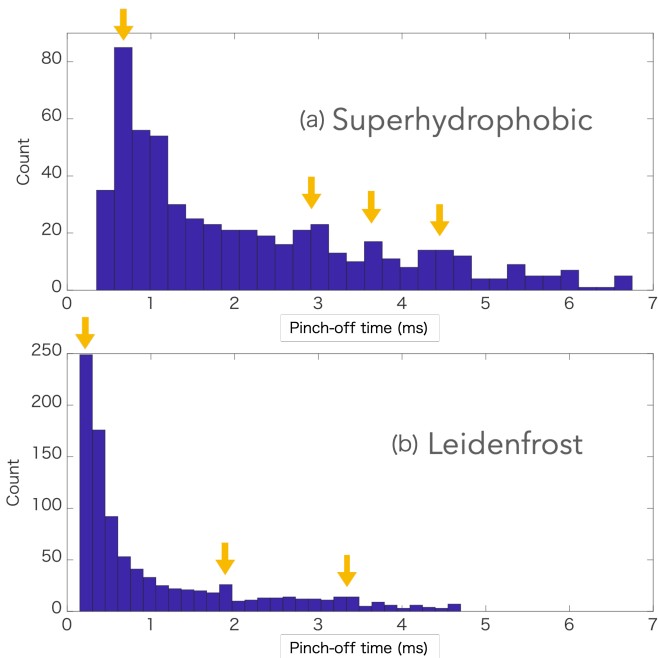

**Figure 9.** Histograms of pinch-off time of daughter drops on a (**a**) superhydrophobic and (**b**) superheated surface.

Let us consider the relation between the daughter drop diameter and the rim diameter. We replot the diameters separately for SH and SHPB surface on Figure 10 as a function of time. This figure shows that the daughter drop diameter is 20 % larger than the rim on the SPHB surface and 40 % on the SH surface.

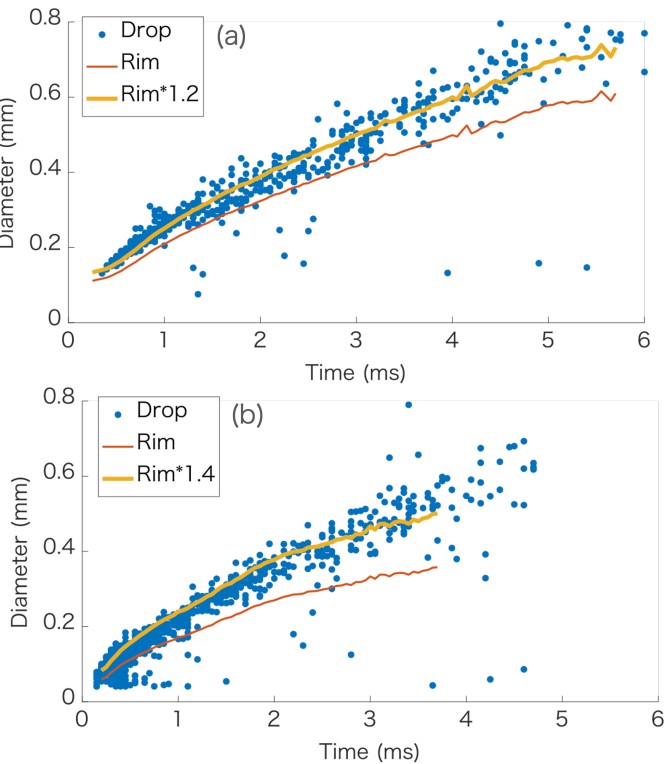

**Figure 10.** Comparison between the diameters of daughter drops (blue circles) and the rim (red lines) on a (**a**) superhydrophobic and (**b**) superheated surface. The yellow lines show $\beta$ times the rim diameter with $\beta$ equal to (**a**) 1.2 and (**b**) 1.4.

## 4. Discussion

We will discuss the effect of vapor flow for the rim breakup on an SH surface. In the previous section, we revealed that the daughter drop size on an SH surface was determined by neither the capillary length nor the rim thickness $b(t)$. This contradicts the breakup on a pillar, where the rim breaks up when its thickness reaches the capillary length, i.e., $Bo = 1$.

Regarding the time scale, we clarified that the daughter drop size is a function of pinch-off time (Figure 8), and on an SH surface, drop pinch off occurs earlier than on an SPHB surface (Figure 9). There remains one more time scale that contributes to determine the drop size: the necking time.

The necking time is required for the capillary bridge between the daughter drop and the rim to be closed. Even during the short period, the surface tension of the film surrounded by the rim decelerates the rim while the ligaments elongated from the rim move at a constant velocity. Therefore, the velocity of the daughter drop at the pinch-off should equal the velocity of the rim at the necking time prior to the pinch-off.

Following the analysis by Wang and Bourouiba [22] and the literature therein, the necking time can be described as

$$t_n = \alpha \sqrt{\frac{\rho w^3}{8\sigma}}, \tag{4}$$

where $w$ is the thickness of the neck, which can be modeled as $w \approx 2/3d$ obtained for the end-pinching drop of a Worthington jet. The pre-factor of the necking time, $\alpha$, slightly differed in prior literature, ranging from 3 to 5.

We look for the necking time from the velocity difference between the daughter drop and the rim. Our experimental results show that the necking time scale, Equation (4), also applies to the drop impact on SH and SHPB surfaces, as shown in Figure 11. Here, the pre-factor $\alpha = 5$ best fitted to both the cases. The vapor flow, therefore, does not contribute to accelerate the necking motion of the capillary bridge between the daughter drop and the rim. In addition, note here that by substituting $\alpha = 5$ and $w = 2/3d$ into Equation (4), we end up with $t_n = \sqrt{\frac{25}{27}\frac{\rho d^3}{\sigma}} \approx \sqrt{\frac{\rho d^3}{\sigma}}$, which corresponds to the capillary time of the drop of diameter $d$.

In summary, we conclude that the vapor flow contributes to enhance the instability of the rim. On an SH surface, the rim starts to destabilize earlier than on a SHPB surface. Once the rim destabilizes, the blobs elongate due to the deceleration of the rim. Because the necking time is proportional to $d^{3/2}$, the earlier destabilization of the rim by the vapor flow leads to the earlier pinch-off, and, thus, generations of smaller daughter drops. On SHPB surface, the vapor flow contribution does not appear. However, due to the contactless spreading between the surface and the rim, the rim breakup conditions obey the closer condition for the impact on a pillar, $Bo = 1$. This finding confirms that the higher $Bo$ of about 2.5 on a normal flat surface [22] was caused by the friction between the rim and the surface during the spreading.

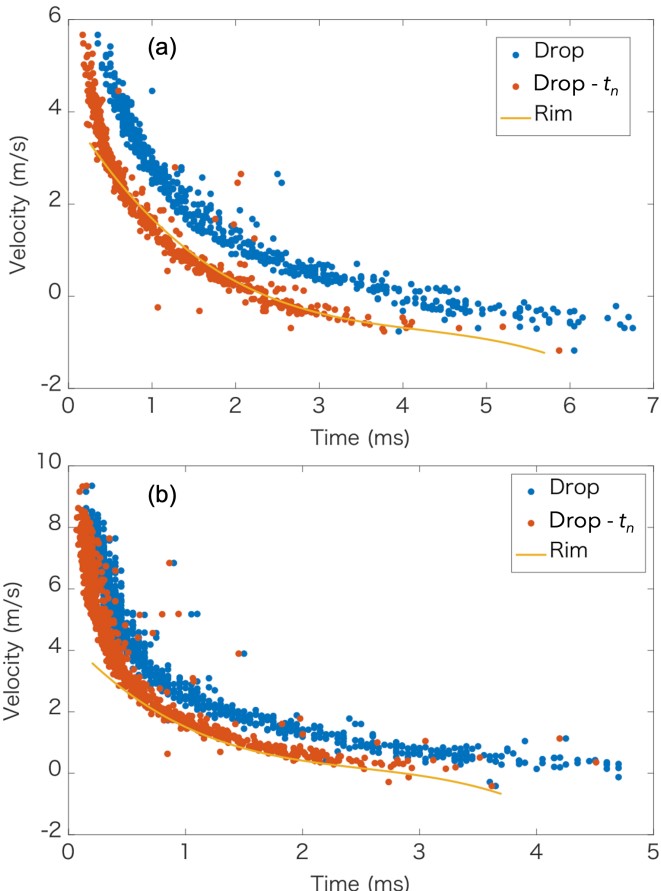

**Figure 11.** Comparison between the velocities of the daughter drops (blue circles) and the rim (yellow lines) at the moment of pinch-off on a (**a**) superheated and (**b**) superhydrophobic surface. The orange circles show the velocities of the daughter drops (blue circles) shifted in time by the corresponding necking time given by Equation (4) with $\alpha = 5$.

**Supplementary Materials:** The following are available online at https://www.mdpi.com/article/10.3390/fluids7020079/s1, Video S1: Extracted images of daughter drops with PTV for the impact on a superheated surface, and Video S2: for the impact on a superhydrophobic surface.

**Author Contributions:** Conceptualization, M.S.; methodology, M.S., M.K., and A.I.; software, M.K. and A.I.; validation, M.S. and M.K.; formal analysis, M.S. and M.K.; investigation, M.K. and A.I.; data curation, M.K. and A.I.; writing—original draft preparation, M.S.; visualization, M.S. and M.K.; project administration, M.S.; funding acquisition, M.S. All authors have read and agreed to the published version of the manuscript.

**Funding:** This research was funded by JSPS KAKENHI Grant Number JP17H03169.

**Data Availability Statement:** The data present in this study are available on request from the corresponding author.

**Conflicts of Interest:** The authors declare no conflicts of interest.

## Abbreviations

The following abbreviations are used in this manuscript:

SH      Superheated
SHPB    Superhydrophobic

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
