# Peer review of "Rim Breakups of Impacting Drops on a Superhydrophobic Surface and a Superheated Surface"

_fluids, doi:10.3390/fluids7020079_

Round 1
Reviewer 1 Report
The article “Rim breakups of impacting drops on a superhydrophobic surface and a superheated surface.” by Shirota et al is reviewed. The article considers the fragmentation of the rim that forms when droplets impact on smooth solid surfaces, with a focus on superhydrophobic and superheated surfaces (above the Leidenfrost temperature).
The article is well-written and describes mainly experimental results in a relevant sub-topic of fluid mechanics. The theoretical analysis lack context (links to previous literature, such as Villermaux and Bossa (JFM2011)). The conceptual analysis of the results (on how the differences between the two surfaces arise) is not clear and would benefit from a sketch.
Still, capturing the presented experimental data is challenging and these provide a valuable contribution to the field, especially since the data have been thoroughly quantified and analyzed.
The following issues, some of which are major, need to be addressed prior to publication:
- It is not clear why the authors remark “…which contradicts to the impact on a pillar where the rim does not exceed an instantaneous capillary length” (line 109). For example, in figure 7b of Villermaux and Bossa (JFM2011), the rim seems to increase in thickness.
- A scheme in which the key parameters R and b are depicted is needed.
- In Visser et al. (Soft Matter 2015), the rim radius was found to scale with t^0.68, and also data by De Ruiter et al were included. Is this scaling recovered?
- What is the meaning of the colors in figure 5? Are these indicating the velocity or moment of ejection?
- It could be interesting to plot figure 7 log-log and see if a power-law dependency emerges (perhaps also uses boxes that contain multiple data points, the overlap is too big!). Note that a ^1/2 scaling was observed for the size of rim corrugations in time for impact on a pillar (Villermaux and Bossa, fig. 8b). Perhaps these results are consistent, if assuming that the drop size scales with the rim corrugations (as indeed observed from fig. 9).
- Do the authors have evidence that impact on SH surfaces is frictionless?
Reviewer 2 Report
This paper presents an original experimental study of the destabilization criterion of a liquid sheet resulting from the impact of a drop on two types of repulsive ( Leidenfrost and super hydrophobic ) surfaces, as well as the size and velocity distribution of daughter droplets emitted from the rim of the sheet. However, it appears to be more of a preliminary study than a finalized article and the writing has many weaknesses. The paper needs a major revision, before being reconsidered with more experiences and a clearer presentation of the methodology and analysis of the results.
Bellow some ( non-exhaustive) comments:
-The experiments are limited to the impact of a drop of water of a single diameter at a single impact speed. This is really insufficient to be able to define in a credible way an experimental criterion of quantitative destabilization in terms of the number of Bond of the rim according to the type of impacted surface. We don't even know if the experiments have been repeated and if so how many times? It would be necessary to consider at least three Weber numbers 3 to 5 times(i.e. three impact speeds) for each type of surface and to evaluate for each of these situations if the quantitative criteria on the number of Bonds (0.5 versus 1.2) determining the destabilization are independent of the Weber number.
- In figure 4a, how are the error bars defined? From the standard deviation of the azimuthal variations of R(t) for a single experiment since the rim is not perfectly circular, or from the standard deviation obtained from the azimuthal means of several impact experiments? Same remark for figure 4c.
- I am surprised and a bit skeptical about the results shown in figure 4b. The acceleration of the rim is obtained by a double experimental derivation of the measurement of R(t) which is itself quite noisy. How then do these curves not show a much higher noise? They appear perfectly smooth... My own experience with this same type of analysis has shown me that this is not the case, and that one must be very careful with the smoothing conditions used for this numerical calculation. In particular, I do not understand that, at least for the Leidenfrost conditions, the acceleration does not have a much flatter maximum, given the shape of R(t) which shows that the velocity hardly varies around the maximum expansion. Consequently, one may wonder whether the non-monotonic variation of the Bond number as a function of time ( figure 4d) is not an artifact of experimental analysis and is not reduced to a simple inflection. The authors should carefully check their experimental analysis procedure.
-Figure 4d is plotted using non dimensional time unit, so it is impossible to compare the evolution of the Bond number with respect to the acceleration and the rim diameter (figure4abc). Moreover, the numerical value of \tau_c should be given.
-Description of the experimental method and drop size analysis for the daughter drop size distribution is totally absent and replaced by an acronym never defined PVT ( particule velocimetry tracking?). The authors should carefully described the methodology . In particularly, in figure 7 here are hundreds of daughter droplets from a single drop of diameter 2.8mm, in addition to a liquid sheet that remains partly intact; and these daughter drops themselves have diameters that can be significant (of the order of 1mm). So the authors must explain very precisely the counting method (combination of repeated trials? ) and the evaluation of the drop diameters as well as the total conservation of the liquid volume.
How the pinch off time versus the breaking time are defined and measured should be reported.
-The authors suppose that the rim section is circular, which is far to be evident ( see for instance Sunderhauf et al Physcis of Fluids, 14, 198 (2002); Comment
-The bibliography should be reviewed, and not be limited to the articles of the Bourouiba and Villermaux groups in this field which are abundantly cited by the authors. The existence of alternative repulsive impact surfaces such as cold Leidenfrost conditions seems to be ignored.
Round 2
Reviewer 1 Report
The manuscript has substantially improved.
However, I would still strongly recommend to include a schematic drawing or sketch of (for example) the cross-section of the spreading droplet that exactly shows how R(t) and b(t) are defined. Putting this in words easily leads to confusion.
Author Response
We thank the reviewer for his/her further suggestion to include a figure illustrating R(t) and b(t). In Figure 2 of the revised manuscript, we have enlarged the image showing the circles fitted to the rim and added arrows showing the definitions of R(t) and b(t). We have accordingly added the explanation in the main text as highlighted on page 3 of the attached manuscript. Again, we thank the reviewer for all the valuable suggestions.

Reviewer 2 Report
The authors have taken into account the comments and questions. The new version is really improved. I rercommand publication in the present form.
Author Response
We thank the reviewer again for all the comments that he/she kindly made to us to improve our original manuscript.